# Role of Adenosine and Purinergic Receptors in Myocardial Infarction: Focus on Different Signal Transduction Pathways

**DOI:** 10.3390/biomedicines9020204

**Published:** 2021-02-18

**Authors:** Maria Cristina Procopio, Rita Lauro, Chiara Nasso, Scipione Carerj, Francesco Squadrito, Alessandra Bitto, Gianluca Di Bella, Antonio Micari, Natasha Irrera, Francesco Costa

**Affiliations:** 1Department of Clinical and Experimental Medicine, University of Messina, 98165 Messina, Italy; mcristinaprocopio@gmail.com (M.C.P.); lrritalauro@gmail.com (R.L.); chiara.nasso@unime.it (C.N.); scarerj@unime.it (S.C.); fsquadrito@unime.it (F.S.); abitto@unime.it (A.B.); gianluca.dibella@tiscali.it (G.D.B.); dottfrancescocosta@gmail.com (F.C.); 2Department of Biomedical and Dental Sciences and Morphological and Functional Imaging, University of Messina, A.O.U. Policlinic “G. Martino”, 98165 Messina, Italy; micariantonio@gmail.com

**Keywords:** myocardial infarction, adenosine, purinergic receptors, P2Y12, Wnt, β-catenin, fibrosis, wnt inhibitors

## Abstract

Myocardial infarction (MI) is a dramatic event often caused by atherosclerotic plaque erosion or rupture and subsequent thrombotic occlusion of a coronary vessel. The low supply of oxygen and nutrients in the infarcted area may result in cardiomyocytes necrosis, replacement of intact myocardium with non-contractile fibrous tissue and left ventricular (LV) function impairment if blood flow is not quickly restored. In this review, we summarized the possible correlation between adenosine system, purinergic system and Wnt/β-catenin pathway and their role in the pathogenesis of cardiac damage following MI. In this context, several pathways are involved and, in particular, the adenosine receptors system shows different interactions between its members and purinergic receptors: their modulation might be effective not only for a normal functional recovery but also for the treatment of heart diseases, thus avoiding fibrosis, reducing infarcted area and limiting scaring. Similarly, it has been shown that Wnt/β catenin pathway is activated following myocardial injury and its unbalanced activation might promote cardiac fibrosis and, consequently, LV systolic function impairment. In this regard, the therapeutic benefits of Wnt inhibitors use were highlighted, thus demonstrating that Wnt/β-catenin pathway might be considered as a therapeutic target to prevent adverse LV remodeling and heart failure following MI.

## 1. Introduction

Myocardial infarction (MI) is one of the main causes of death worldwide [1]. Several processes are activated following MI and in particular those associated with tissue remodeling; for this reason, MI may represent a dramatic event, mostly when it is related to adverse cardiac remodeling and deterioration of ventricular fraction [2,3,4]. Left ventricular (LV) remodeling is an adaptive response that is activated in order to preserve heart wall structure; however, the excessive LV remodeling may degenerate into a maladaptive phenomenon up to heart failure [2,4]. The “first line” therapeutic approach used in patients following an acute myocardial infarction is a rapid reperfusion of the affected vessels by the percutaneous coronary intervention (PCI). After revascularization, different therapeutic strategies are applied to cope with the subsequent phases post-PCI in order to avoid MI progression, such as beta-blockers, renin-angiotensin system inhibitors, antiplatelet drugs and statins [5,6]. However, despite these different therapeutic options, an irreversible and pathological remodeling may be triggered, as a consequence of the activation of several cellular and molecular pathways [7]. In the last years, a role of the Wnt/β catenin signaling has been demonstrated in myocardial dysfunction and its stimulation may be involved in the myocardial diseases, especially those characterized by fibrosis, thus exacerbating cardiac damage and worsening the left ventricular systolic function [6,8,9,10,11,12,13,14,15,16,17,18].

In this context, the adenosine and purinergic systems play a fundamental role not only in the development and in the normal physiology of the cardiovascular system [19,20,21,22,23,24,25] but also in heart diseases [26,27,28,29,30], also through Wnt/β catenin pathway involvement. In the light of these evidences, both the modulation of the adenosine pathway and of the purinergic system might be effective for the treatment of heart diseases, thus avoiding fibrosis, reducing infarcted area, limiting scaring and promoting functional recovery.

## 2. Purinergic Receptors

Purinergic receptors, also known as Purinoceptors, are transmembrane receptors for both adenosine and extracellular nucleotides, such as ATP, ADP, UTP and UDP [31].

Purinergic receptors either belong to the class of GPCRs or Ligand-gated Ion Channels. They are often distinguished into two major families depending on their ligands: P_1_ receptors, stimulated by adenosine, and P2 receptors, stimulated by the other extracellular nucleotides. In accordance with their receptor class, all P1 receptors—which are usually referred to as A_1_R, A_2A_R, A_2B_R and A_3_R—as well as some P2 receptors, being GPCRs, are named P1Y and P2Y. The remaining part of P2 receptors, belonging to the other class of receptors, are named P2X [32,33].

These metabotropic receptors are widely distributed in different tissues: brain, adipose tissue, kidneys and heart [34].

ATP, the main and most representative extracellular nucleotide, can be released from endothelial cells, red blood cells (RBCs), platelets activated in presence of vascular injury, inflammatory cells and smooth muscle cells following hypoxic events, acidosis or adrenaline in the perfused heart [35,36,37]. As mentioned, ATP may bind P2 receptors.

Despite of the interaction with a variety of effector proteins and/or with organic ion influx, P2 receptors activation may increase intracellular Ca^2+^ and stimulate the G proteins RhoA and Rac. The increase of cytosolic Ca^2+^ concentration can be mediated by both the opening of Ca^2+^ channels or the activation of PLCβ by G_q_ proteins [38].

The activation of the monomeric G proteins RhoA and Rac leads, on the other hand, to cytoskeletal rearrangements involved in cell migration, and it has recently been discovered their contribution to cardiac fibrosis induced by pressure overload [39].

The purinergic signaling is an important regulator of the heart rate, contractility and coronary flow. 

P1Y receptors, expressed on cardiomyocytes [40], promote negative chronotropic and dromotropic effects, specially A_1_R, which can also cause a prolongation of PR interval until atrio-ventricular block, whereas A_2A_R enhances cardiomyocytes contractility and shortening [41]. ATP, instead, when bound to P2 receptors, promotes a dual effect in mammalian hearts: it can enhance either a positive or a negative ionotropic effect. Indeed, after a period of cardiac arrest, it has been shown that ATP had a strongly positive ionotropic effect; however, mechanism of P2 receptors desensitization results in a negative ionotropic effect which can occur with increased ATP levels, thus weakening the force of muscular contractions of cardiomyocytes and inducing a dysfunction in myocardium [42,43]. Regarding the chronotropic effect, the effect of ATP is dose-dependent: small concentration of ATP induces tachycardia, while a higher concentration promotes bradycardia, probably due to ATP degradation to adenosine [44]. Moreover, ATP also does not induce cell hypertrophy: even if the intracellular signaling induces the activation of molecules involved in cell growth, their levels are not sufficient to stimulate hypertrophy [45].

Purinergic signaling regulates the vascular tone in the endothelium: adenosine, AMP, ADP and ATP induce vasodilatation [46,47], mainly acting on P2 receptors and A_2A_R and A_2B_R. Their signaling mechanisms, indeed, promote the synthesis of NO, endothelium-derived hyperpolarizing factor and/or prostaglandins [48,49]. However, in an injured endothelium, ATP may act as a vasoconstrictor, because of its binding with P2Y4 and P2Y6 receptors localized on vascular smooth muscle, leading to local vasospasm [50].

On both endothelial cells and vascular smooth muscle cells, P1 and P2 receptors signaling may mediate their proliferation. Specifically, P1 receptors stimulate endothelial cell proliferation but inhibit smooth muscle cell proliferation, whereas P2 receptors stimulate both endothelial and smooth muscle cell proliferation, because of the activation of mitogenic effectors [51,52,53,54].

At the platelets level, the effect that the purinergic system exerts results in an increased aggregation. Among P2 receptors, P2Y_1_, and P2Y_12_ are principally located on their surface; both receptors, particularly P2Y_12_, are involved in platelet aggregation [55]. In fact, in normal conditions, platelets don’t make contact with undamaged vessels wall. However, when a vascular injury occurs, platelets interact with components of the subendothelial extracellular matrix, aiming to block the bleeding and promoting tissue repair. Many components of the extracellular matrix—as collagen, laminin, fibronectin, von Willebrand factor—bind their respective receptors on the platelet surface, resulting in platelets adhesion and activation [56]. Specifically, once platelets are activated, different signaling pathways are stimulated. The main detected effects are cytosolic Ca^2+^ concentration increase, changes in the platelets’ shape, degranulation and finally platelet aggregation. After granule secretion, ADP is released from dense granules in the extracellular environment and can bind its purinergic receptors on the platelet membrane, resulting in a positive feedback loop to the platelet aggregation and arterial thrombus formation, hallmarks in the pathogenesis of MI [57]. Therefore, playing a key role in hemostasis and thrombosis, P2Y_12_ is currently considered as a therapeutic target for the management and prevention of arterial thrombosis [30].

Nevertheless, both A_2A_R and A_2B_R receptors are expressed in platelets: ARs activation increases intracellular cAMP, thus reducing platelet activation and aggregation through the increase of cAMP [58]. Therefore, targeting ARs might be considered as a therapeutic approach for the management of thromboembolic events, as an anti-platelet therapy.

It has also been shown the role of purinergic receptors in the induction of fibrosis. Adenosine, mainly through A_2B_R, also inhibits collagen and protein synthesis in cardiac fibroblasts. However, A2AR activates fibroblasts to produce collagen I (Col1), collagen III (Col3) and to modulate matrix metalloproteinase (MMP) 9,2 and 14 [59]. Even extracellular nucleotides, especially UTP, binding P2Y2 receptors, induce pro-fibrotic responses in rat and mouse cardiac fibroblasts [60]. ATP, in particular, via P2Y2, PX2SR and P2X7R, up-regulates the proliferation and migration of human cardiac fibroblasts, promoting cardiac fibrosis [61].

There are conflicting reports regarding the actions of the purinergic signaling during MI. Generally, the role of these purine and pyrimidine nucleotides is considered protective, when released after an injury of the heart. This is mostly because adenosine is considered as a potent vasodilator in reactive hyperemia following hypoxia. However, since their action it is likely exerted in the phase of post-infarct remodeling, due to the induction of fibrosis, the modulation of purinergic receptors could be considered protective against cardiac fibrosis and may also regulate cardiac remodeling following MI [62].

## 3. Adenosine Receptors

Adenosine is an endogenous nucleoside released by the cells as a consequence of both physiological and pathological conditions [63,64].

Since adenosine is involved in the normal development of the cardiovascular system, as well as in the organ function regulation, it has been and it is currently considered as a diagnostic and therapeutic tool in the cardiology field [64]. Specifically, adenosine influences the rate and the strength of beating, regulates the generation and conduction of the cardiac impulse, is involved in coronary perfusion and vascular control, modulates myocardial metabolism and cardiac contractility [63,65].

Adenosine is a purine nucleoside mainly produced by the dephosphorylation of nucleotides such as Adenosine Triphosphate (ATP), Adenosine Diphosphate and Adenosine Monophosphate (AMP) both in the extracellular and intracellular compartments. Extracellular adenosine is quickly put in the red blood cells through the action of equilibrative nucleoside transporters (ENT); here, it is metabolized in inosine by adenosine deaminase (ADA) or converted in adenine nucleotide by adenosine kinase; for this reason, extracellular adenosine has a short half-life, which can be protracted through ENT inhibition [66]. The main enzymes involved in the production of adenosine belong to the class of nucleotidases: ecto-apyrase (CD39) and ecto-5′-nucleotidases (CD73) [58,67]. Adenosine can also originate from S-Adenosyl-Homocysteine through the action of the S-Adenosyl-Homocysteine hydrolase [68], that could explain the increase of plasma levels of adenosine and homocysteine in some conditions. Moreover, a soluble form of 5′ nucleotidases, which is released with ATP under the stimulus of sympathetic nervous system, is involved in the production of adenosine from ATP [69]. Adenosine carries out its downstream effects by binding to specific receptors named Adenosine Receptors (ARs) [65]. Four subtypes of ARs can be distinguished—A_1_R, A_2A_R, A_2B_R, and A_3_R—depending on their different response consequent to their activation; moreover, A_1_R and A_2A_R showed a higher affinity for adenosine compared to A_2B_R and A_3_R, which have a lowest affinity [70].

ARs belong to the class of G-Protein Coupled Receptor (GPCRs). In particular, A_1_R and A_3_R are coupled with G-inhibitory (G_i/o_) proteins, while A_2A_R and A_2B_R are coupled with G-stimulatory (G_s_) proteins.

The signaling induced by A_1_R and A_3_R inhibits the activation of the membrane-bound protein Adenylate Cyclase (AC), which catalyzes the conversion of ATP to 3’,5’-cyclic AMP (cAMP). On the other hand, A_2A_R and A_2B_R signaling is characterized by the increase of intracellular cAMP levels, due to the activation of AC. cAMP is responsible for the activation of three main targets: the serine/threonine Protein Kinase A (PKA), the Exchange protein activated by cAMP (Epac) and Cyclic Nucleotide-Gated Ion Channels (CNGCs) [65]. PKA catalyzes the phosphorylation of several cellular substrates, acting on both ion channels, cellular transporters and exchangers, Ca^2+^ channels and pumps, and Transcription Factors (TFs), thus conferring it a considerable role in the regulation of gene transcription and in the influence of the heart contractility [71]. In fact, ARs are ubiquitously distributed in different organs and tissues and also in heart tissue [72] and in coronary arteries [73]; it has been shown that PKA dysfunction is related to cardiac dysfunction and heart failure [74].

On the other hand, Epac proteins, as a guanine nucleotide exchange factor (GEF), act on Rap1 and Rap2, small G-proteins known to control endothelial barrier resistance and angiogenesis through the regulation of cell adhesion, proliferation and cell migration [75].

Lastly, CNGCs activation is crucial to ensure the optimum cardiac pacemaking and ventricular repolarization through the transition of positively charged ions [76].

Comprehensively, the activation of ARs by endogenous and exogenous adenosine is responsible for both the induced and endothelial-independent vasodilatation [21], an effect mainly mediated by the cAMP dependent pathway enhanced by A_2A_R and A_2B_R, and vasoconstriction through A_1_R and A_3_R [74,77].

However, there are other signaling pathways induced by the activation of the βλ subunits of G proteins, either G_i/o_ or G_s_ proteins.

Phospholipase Cβ (PLCβ), for example, activated by such subunits, is responsible for the activation of the Protein Kinase C (PKC) [65]. PKC has been reported to have a major role in phosphorylating and regulating the activity of many TFs which control the expression of hypertrophic genes [78]. For this reason, PLC-induced pathway can be considered determinant of cardiac hypertrophy and its inhibition could represent a potential approach for cardioprotection.

Phosphatidyl-Inositol-3-Kinase/AKT (PI3K/AKT), extracellular signal-regulated kinase (ERK) and Jun NH2-terminal kinase (JNK) signaling are other transduction pathways regulated by the activation of ARs, and predominantly by A_1_R (Figure 1). Their downstream responses include adenosine-induced mitogenesis [79].

ARs furtherly extrinsic their actions influencing K^+^ channels activity, especially K_ATP_ channels. In particular, ARs seem to induce their opening, leading to Ca^2+^ influx and the subsequent increase of intracellular Ca^2+^. The positively charged ion, binding to many substrates, calmodulin in particular, may directly activate Nitric Oxide (NO) synthesis [80].

All of the cellular effects provided by adenosine further strengthen ARs’ involvement in mechanisms of vascular regulation, and they give evidences of their critical role in the development of left ventricular dysfunction as well.

Adenosine receptors are commonly considered as independent receptors, however, a cross-talk may be induced through heterodimerization with other GPCRs. In particular, A1R may interact with P2Y1 or D1 dopamine receptors [81,82], whereas A2AR may heterodimerize with P2Y, D2 dopamine and mGLU5 receptors [83,84,85]. Therefore, the AR system shows different interactions between its members and with other GPCRs and this makes sense since purinergic receptors represent one of the earliest signaling systems involved in the regulation of cellular function and energy state [86].

## 4. Wnt/β-Catenin Signaling Pathway Role in Myocardial Infarction (MI)

The Wnt signaling represents a complex network involved in cell growth and cell proliferation, essential in embryogenic development processes. However, an aberrant Wnt signaling stimulation might be crucial in developing degenerative diseases and cancer [87].

Wnt pathway is highly regulated both temporally and spatially during development and is differently modulated depending on age, cell types and different tissues. Wnt glycoproteins act as ligands of specific receptor members belonging to the Frizzled (Fz) receptor family; Wnt pathway activation also requires the engagement of co-receptors, such as the low-density-lipoprotein-related protein 5/6 (LRP5/6) [88]. Wnt signal stimulates different cascades, including (a) the canonical Wnt/β-catenin signaling pathway, (b) the non-canonical Wnt/Planar Cell Polarity (PCP) pathway and (c) WNT/Ca^2+^ pathway [89]. Several Wnt glycoproteins are recognized which may induce both the canonical and the non-canonical Wnt pathway activation [87]. The canonical Wnt pathway, which is also the best known, is a β-catenin-dependent signaling.

β-catenin is a protein which regulates cellular transcription and gene expression. Without Wnt signaling, it is localized in the cytoplasm, bound and “blocked” by a complex, called β-catenin destruction complex, made of the following proteins: Axin, Adenomatosis Polyposis Coli (APC), Protein Phosphatase 2A (PP2A), Glycogen Synthase Kinase 3 (GSK3) and Casein Kinase 1α (CK1α) [90]. Specifically, β-catenin is susceptible to sequential phases of phosphorylation regulated by the above-mentioned proteins. The phosphorylated β-catenin can be recognized by a ubiquitin ligase complex, called β-Trasducin. The ubiquitination of β-catenin promotes its degradation by the proteasome [9,91]. In this way, the cytoplasmatic and especially β-catenin nuclear concentration is kept low in absence of Wnt ligands [92].

The activation of the complex composed by Fz receptor and the LRP/6 co-receptor induces β-catenin translocation into the nucleus; here, β-catenin interacts with different transcription factors (members of the TCF/LEF family) and co-activators (p300 and/or CREB-Binding Protein—CBP), thus activating Wnt target genes transcription (Figure 2) [93,94]. In cardiac tissue, Wnt proteins expression involved in the canonical pathway modulation is significantly reduced in postnatal heart, probably due to the fulfilment of heart development. However, scientific evidences demonstrated that the activation of the canonical Wnt pathway may play an important role on the expansion of cardiac progenitors after cardiac stress and vascular damage. Therefore, its inhibition has been considered cardioprotective, as it would inhibit fibrotic processes induced after a cardiac damage [95]. In addition to Wnt ligands, other molecules can regulate β-catenin translocation, such as Akt [96]. Akt activation can be triggered by adenosine, an activator of the canonical Wnt pathway: A_2A_R stimulation by adenosine increases cAMP levels which can activate PKA. PKA, in turn, mediates Akt phosphorylation and activation. The β-catenin destruction complex, and particularly GSK3, can be phosphorylated and subsequently inactivated by Akt [97]. This phenomenon may induce β-catenin translocation into the nucleus. Akt activation can be also promoted by the PI3K pathway, which can be induced by the activation of A_2A_R, A_2B_R and P2Y [98,99]. Specifically, phospholipids phosphorylated by PI3K can recruit Phospholipids-dependent kinase 1 (PDK1), which can subsequently activate Akt [100]. Coexisting with the canonical pathway, also the non-canonical Wnt pathways, known as *β-*catenin-independent pathways, may play important roles. In these pathways, β-catenin translocation is not involved, but Wnt canonical pathway inhibition is carried out in order to reduce the overall concentration of β-catenin. In particular, in the Wnt/PCP pathway, the Dishvelled (Dvl) protein is recruited following Wnt interaction with Fz receptor independently of co-receptors LRP5/6. Dvl is responsible for the activation of small GTPases such as RHOA (Ras homolog gene family, member A), RAC (Ras-related C3 botulinum toxin substrate) and Cdc42 (cell division control protein 42). The cellular effects of their activation are integrated for regulation of actin cytoskeleton, cytoskeletal rearrangements and changes in cell polarization and/or directed migration [87,101].

Lastly, the Wnt/Ca^2+^ pathway regulates Ca^2+^ release, which may be essential in the regulation of cardiac contraction. In fact, the activation of some G-proteins could stimulate the release of intracellular Ca^2+^ from the Endoplasmic Reticulum (ER). Ror/Ryk co-receptors, forming a complex with Fz receptor, when bound by a specific ligand, recruit Dvl protein. Dvl subsequently induces trimeric G-proteins activation, responsible for PLCβ-triggered release of Ca^2+^. Ca^2+^, associating to calmodulin, stimulates the activation of Calcium/calmodulin-dependent kinase II (CAMKII) [102], TGF-β Activated Kinase 1 (TAK-1) [102] and Nemo-Like Kinase (NLK) [103]. Their activation further inhibits Wnt canonical pathway and stimulates calcineurin and the transcriptional factor Nuclear Factor of Activated T-cells (NFAT), leading to the expression of several genes that are involved in cytoskeletal remodeling processes in different tissues, including heart [104].

In the heart, non-canonical Wnt pathways might antagonize canonical Wnt signaling during cardiac development, and this would explain the sophisticated balance between the non-canonical and the canonical Wnt pathway. Indeed, Wnt signaling pathways are activated during cardiomyogenesis, although they are consequently inhibited to avoid an excessive heart development [105]; however, a further activation of canonical Wnt pathway might be enhanced following myocardial injury [8], due to mechanisms of “counter-regulation” of the infarct process in order to preserve the heart attacked area. However, the activation of this pathway in a post-infarct environment could contribute to the establishment of LV remodeling. An excessive canonical Wnt activation might activate signaling cascades, promoting rearrangements of the cytoskeletal structure of cardiomyocytes. This final effect could potentially lead to cardiac remodeling and let us to hypothesize that abnormalities of these pathways might be considered as risk factors for the development of different cardiovascular pathological conditions. Therefore, the fine regulation of Wnt signaling pathways is essential in repairing and protecting cardiac tissue post MI [106]. As a matter of fact, several evidences showed a correlation between myocardial injury and Wnt signaling pathway activation.

Three main phases are widely recognized following MI: inflammation, angiogenesis and fibrosis [7,107]. Inflammatory cells stimulate cytokines and chemokines release; under the stimulus of chemokines, circulating monocytes reach into the infarcted myocardium zone and here differentiate into macrophages. Macrophages are responsible for the removal of apoptotic and necrotic cardiomyocytes and for the repair of damaged tissue [108,109,110]. After the removal of cell debris, angiogenesis is activated and myofibroblasts start to produce collagen to replace degenerated cardiomyocytes. A recent study revealed that macrophages do not express active β-catenin in healthy heart and Wnt signaling activity is held in normal conditions [111]. On the other hand, some evidences indicated that Wnt ligands are upregulated after myocardial infarction [112], thus suggesting that β-catenin is crucial for macrophages-mediated inflammatory reaction [113] and may promote phagocytic activity. The tissue damage consequent to MI stimulates the release of several molecules involved in fibrotic and inflammatory processes, such as Transforming Growth Factor-β1 (TGF-β1), activated Nuclear Factor-κB (NF-κB) and inflammatory cytokines [114,115]. Both TGF- β1, NF-κB and pro-inflammatory cytokines may contribute to Wnt/β-catenin pathway activation [9,10,14,18], thus promoting β-catenin translocation. In addition, they could also induce the release of ATP metabolites and adenosine. Despite of their defense role, they may indirectly stimulate the canonical Wnt signaling pathway. For instance, a great amount of adenosine is provided from cardiac injury because of its exportation from apoptotic and necrotic cells. All these events may exacerbate Wnt/β-catenin signaling and may interfere with the balanced cardiac remodeling, worsening the tissue damage and causing heart failure. Based on these data, cardiac remodeling could be harmful in the infarcted zone and in the remote myocardium, through a positive feedback involving β-catenin. Therefore, the early inhibition of Wnt/β-catenin signaling pathway in the first days after MI could prevent adverse cardiac remodeling and improve LV function, with positive effect on long-term outcome.

## 5. Adenosine Agonists and Antagonists

Endogenous adenosine is a purine nucleoside that controls the balance of different stressful conditions. As a matter of fact, adenosine has been shown to be critical in cardioprotection, thus modulating heart rate and endothelial vasoconstriction, through A_1_R and A_3_R activation, or stimulating blood supply and vasodilatation, through A_2A_R and A_2B_R. In particular, it is extensively known that its dynamic effects can be exerted on the ischemic and/or reperfused heart, with the reduction of the infarct size [40,116]. However, adenosine activity could represent a risk factor during LV remodeling following MI, since Ars activation could stimulate the expression of those genes involved in hypertrophy, including the expression of the MAPKs ERK and JNK that induce mitogenesis.

Nevertheless, adenosine agonists and antagonists have been extensively used to modulate the adenosinergic signaling for the treatment of the damage in the ischemic and/or reperfused heart. However, most of the compounds have not been approved for human use, even if new promising designs are under application for preclinical and clinical approval. At this stage, only the following compounds have been used in clinical trials: A_1_ agonists Selodenoson (DTI-0009) and Capadenoson (BAY68-4986) for ventricular rate control in atrial fibrillation, the partial A_1_ agonist Neladenoson bialanate (BAY1067197) for chronic heart failure with reduced ejection fraction, and the A_2A_ agonist Regadenoson, which is also the only one approved for the treatment of heart failure in humans [117,118]. A_1_R and A_3_R agonists are the most studied and it has been suggested that the simultaneous administration could be beneficial for cardioprotection [119]. In addition, A_2A_R and A2BR modulation may exert different time-dependent and opposite effects in the post-ischemic heart, but a therapeutic effect has been recognized on post-MI remodeling, particularly with the A_2B_R antagonism [120].

Some of the effects mediated by adenosine signaling modulation through adenosine agonists and antagonists use are described below.

### 5.1. A_1_R

A_1_R modulation has offered potential therapeutic benefits for heart failure through the improvement of cardiac functions. However, there are some limitations for the use of full AR agonists, due to the negative ionotropic and chronotropic effects mediated by A_1_Rs [121,122] For example, Trabodenoson (INO-8875, PJ-875), a A_1_R partial agonist, was initially considered for the treatment of open-angle glaucoma and ocular hypertension, but the clinical trials failed and terminated in 2017 [123]; nevertheless, it has been considered for arrhytmias [124] in association with the A_1_R full agonists Tecadenoson (CVT-510) [125] and Selodenoson (DTI-0009) [126], whose chronic use resulted in adverse events due to the receptor desensitizazion [127]. Moreover, the non-nucleoside A_1_R agonist Capadenoson (BAY 68-4986) [128] and Neladenoson bialanate (BAY 10671971) [129] showed anti-ischemic effect in response to heart failure with reduced ejection fraction. BAY 10671971 has also been tested in patients with chronic heart failure and its use did not cause the appearance of undesirable side effects [130,131,132] VCP746 is another A_1_R agonist known for its cytoprotective properties. However, Chuo et al. demonstrated that VCP746 mediated anti-hypertrophic and anti-inflammatory effects in neonatal rat cardiac myocytes [133]. Later, a binding affinity of VCP746 on A2BR has been shown, resulting in a significant anti-fibrotic response [134]. VCP28 is another A2BR agonist which inhibited cell death and increased functional recovery in the post-ischemic cells, thus playing a cardioprotective role [135].

A1R antagonists have been and are currently considered for the treatment of both acute and chronic heart failure [136,137,138,139], thanks to the possible inhibition of vasoconstriction through A_1_R modulation. However, the results obtained by a large and randomized trial of phase 3 (PROTECT), that studied the effects of the antagonist Rolofyllin, did not confirm the initial hypothesis.

### 5.2. A_2A_R

A_2A_R modulation has always been a controversial move for a cardioprotective effect. As a matter of fact, there is currently a poor number of A_2A_R agonists studied for the treatment of heart diseases. The synthetic compound ATL 313 reduced myocardial infarct size and promoted LV contractility in reperfused myocardium but also promoted a severe systemic hypotension thus increasing the mortality rate related to myocardial infarction. Also CGS-21680, acting as A_2A_R agonist, caused hypotension and, in addition, did not reduce myocardial infarct size [140].

However, LASSBio-294 is an A_2A_R agonist that showed a positive effect on heart and significantly reduced hypertension associated with MI and decreased collagen deposition and TNFα levels in the left ventricle, thus inhibiting fibrosis [141]. Therefore, also the modulation of A2AR might be considered as a possible target for the treatment of heart diseases but additional studies will be needed to confirm the hypothesis.

### 5.3. A_2B_R

Previous studies have demonstrated that A_2B_R agonism may be useful for the treatment of the acute phase of myocardial ischemia whereas the A_2B_R antagonism may be considered as a therapeutic approach for chronic diseases, cardiac remodelling and fibrosis [118,142]. Specifically, A_2B_R blockade attenuated cardiac enlargement and lead to a more effective cardiac remodeling by reducing caspase-1 activity, a key mediator that regulates inflammatory response [120]. In addition, GS 6201 ameliorated ventricular dysfunction and decreased fibrosis especially in the border areas of the MI in an in vivo rat myocardial ischaemia reperfusion model [143]. Instead, the use of A_2B_R agonists may inhibit endothelin-1 (ET-1), a potent vasoconstrictor involved in the pathogenesis of cardiac fibrosis. In fact, the treatment with CV1808, a selective A2 receptor agonist, inhibited ET-1-induced fibroblast proliferation [144].

BAY 60-6583, instead, is a compound that may play different roles [145]; it has been firstly considered as an agonist but a role as antagonist has been claimed. This different activity could be due to the several signaling pathways activated following receptor stimulation, whose complexity could lead to different outcomes [146]. BAY 60-6583 promoted protective effects in a myocardial reperfusion injury, thus inducing the shift of macrophages to a M2 phenotype [147].

In addition, the use of GS-6201, as an A_2B_R antagonist, produced better post-infarction heart function [120] although it still led to the formation of a thicker scar, similarly to the results obtained in a chronic occlusion and reperfusion experimental model [143].

### 5.4. A_3_R

IB-MECA and CI-IB-MECA are two prototypical A3R agonists used for the treatment of inflammatory, ophthalmic and liver diseases [148] and their use might promote the recovery of post-ischemic LVDP, thus reducing the infarct size [149].

The protective role of the Cl-IB-MECA agonist against stress conditions (i.e., hypoxia) has been demonstrated in isolated rat cardiac myocytes, whereas the use of an A3R antagonist, MRS1523, reverted the positive effects of the agonist [150]. However, the cardioprotection significantly diminished when Cl-IB-MECA was used at concentrations ≥ 10 μM, increasing the risk of toxicity [151,152].

Also the use of two others selective A3R agonists, CP-608,039 and CP-532,903, may protect cardiomyocytes against an ischemic injury [118,153,154] and in particular, the ischemic tolerance promoted by CP-532,903 seems to be partially due to its role in facilitating the opening of myocardial K_ATP_ channels.

In view of the data discussed so far, the development and the introduction of AR ligands as new drugs for the treatment of heart diseases may be an important challenge in the near future. The efforts of the scientific community aimed at understanding the complex pathways involved in the agonism and antagonism of adenosine receptors in heart will be needed to confirm the promising therapeutic effects.

## 6. The Therapeutic Benefits of Wnt Pathway Inhibitors

Recent findings have drawn attention on the possible application of different compounds, some of which are anti-cancer drugs that could be used as therapeutic strategy in patients after MI to prevent heart failure. These compounds are able to inhibit Wnt signaling pathway and play a cardioprotective role by limiting loss of cardiomyocytes and fibrosis, promoting myocardial regeneration and preventing adverse cardiac remodeling. Insulin-like growth factor binding protein 4 (IGFBP-4) and Dkk1 are two inhibitors of the canonical Wnt/β-catenin signaling pathway [43,44] but play an opposite role in ischemic injury: Dkk1 increases ischemic injury by promoting the endocytosis of basal LRP4/5 and its elimination with a mechanism independent of β-catenin, as demonstrated by in vivo studies [155,156]; instead, IGFBP-4 contrasts ischemia-induced DNA damage by inhibiting LRP5/6 and reducing β-catenin activation both in vitro and in vivo, and only in presence of Wnt proteins and not in basal conditions. Based on these evidences, IGFBP-4 could be a promising drug to prevent myocardial DNA damage after MI [157]. CGX1321, WNT-974 and GNF-6231 are three porcupine inhibitors: the first one inhibits both canonical and non-canonical Wnt signaling pathways, by avoiding Wnt proteins secretion. Recent studies have shed more light on the protective role of this drug on the post-MI heart; in fact, CGX1321 might reduce infarct size and fibrotic zone, contrasting ventricular chamber dilatation and worsening cardiac function in post-MI mice hearts [158]. Moreover, previous studies suggested that the proliferation of pre-existing cardiomyocytes is dominant by comparison with newly formed cells [159,160,161,162]; in this regard, CGX1321 has been shown to increase cardiomyocytes proliferation in vitro, preventing cardiac remodeling [158]. Using in vivo and in vitro approaches, it could be proven that WNT-974 blocks the secretion of Wnt3, a profibrotic compound, in post-MI heart, thus reducing scarring and promoting cardiac function recovery. Furthermore, WNT-974 prevents the phosphorylation on Dvl proteins, acting on both canonical and non-canonical Wnt signaling pathway [163]. Finally, GNF-623, another porcupine inhibitor, limits the adverse cardiac remodeling by reducing extracellular matrix deposition and cardiomyocytes death but stimulating cellular proliferation [164].

Porcupine inhibitors are also administered in clinical trials for the management of advanced tumors.

In particular, WNT974, ETC-159, RXC004 and CGX1321 are in “recruiting state” in phase I trials (NCT01351103, NCT02521844, NCT03447470 and NCT02675946, respectively) even if side effects on bone related to their use have already been registered. Wnt inhibitors use, in fact, might impair bone mass, thus increasing bone resorption: this is why Wnt inhibitors should be administered in association with bone protective agents [165].

Wnt pathway inhibition may also occur at receptor level; in this context, Vantictumab and Ipafricept were used in clinical trials as single agents or in association with anticancer drugs (Paclitaxel, Carboplatin, or Gemcitabine) but also Vantictumab and Ipafricept significantly impaired bone metabolism, thus causing bone fractures [166,167].

Pyrvinium is a Wnt inhibitor, which acts by inhibiting Actin degradation and, similarly to CGX1321, promotes cellular proliferation in the peri-infarcted zone, thus encouraging tissue regeneration [168]. UM206 is a peptide with high affinity for Frizzled-1 and Frizzled-2 family members. It uses the same Wnt3a and Wnt5a binding sites on Frizzled 1 and Frizzled-2 receptors and thus inhibits the downstream effects of both ligands. Wnt3a is involved in the canonical WNT/β catenin signaling, whereas Wnt5a should be involved in the non-canonical Wnt signaling. A recent in vivo and in vitro study demonstrated that UM206 could reduce collagen production and increase the neoformation of blood vessels and myofibroblast proliferation in the infarcted area. Moreover, UM206 could reduce infarct size and could preserve cardiac function after MI injury, by preventing left ventricular dilatation and, as a consequence, could avoid the development of heart failure [169].

Wnt inhibitors use as a therapeutic strategy is supported by a robust rationale, however, the potential clinical application needs to be deeply evaluated: further studies should investigate the sophisticated interaction of Wnt signaling with other pathways and the side effects/toxicity related to their use (i.e., bone remodeling alterations).

## 7. Clinical Outlook–Clinical Studies for Adenosine Receptor Agonists/Antagonists in Humans with Acute Myocardial Infarction

The role of adenosine and adenosine receptors in the clinical setting of acute myocardial infarction (AMI) has been widely studied.

Pre-clinical studies suggested multiple beneficial effects of adenosine agonists use, supporting adenosine as a potential cardioprotective agent during AMI (Figure 3) [170,171]. Treatment with adenosine consistently reduced infarct size, improved left ventricular function and coronary blood flow in animal models: the mechanisms proposed to demonstrate adenosine efficacy are reduction of (i) neutrophil activation and migration, (ii) neutrophil-mediated reperfusion injury, (iii) oxygen free radical formation, (iv) cardiomyocytes apoptosis and increase of coronary blood flow. In addition, adenosine appeared to mimic and potentiate ischemic pre-conditioning, protecting from subsequent reperfusion injury.

Adjunctive adenosine therapy for the management of AMI has been studied in two pivotal clinical trials in humans: Acute Myocardial Infarction STudy of Adenosine (AMISTAD) and AMISTAD II.

AMISTAD was a prospective, open-label trial including 236 AMI patients undergoing thrombolysis and randomized within 6 h of infarction onset to a 3-h intravenous infusion of 70 mg/kg/min of adenosine or placebo [172]. The study ultimately showed a 33% relative reduction in infarct size (*p* = 0.03) with adenosine in the overall population, with the highest reduction of infarct size among patients presenting anterior AMI (67% relative reduction in infarct size, 15% in the adenosine group vs. 45.5% in the placebo group), whereas no difference was observed among patients with nonanterior AMIs (infarct size 11.5% in both groups). There was no significant difference in clinical outcomes between treatment groups, but the overall number of events was small and the trial was underpowered to evaluate any clinical benefit. Hence, to better understand the impact of adenosine therapy on clinical endpoints the larger AMISTAD-II trial has been designed [173].

AMISTAD II was a double-blinded, placebo-controlled, randomized study which enrolled 2,118 patients within 6 h of an anterior AMI undergoing thrombolysis or primary angioplasty. Patients included were randomized in a 1:1:1 fashion to a 3-h infusion of either low-dose (50 µg/kg/min) or high-dose (70 µg/kg/min) adenosine or placebo. The primary end point was new congestive heart failure (CHF) 24 h or more after inclusion, or occurrence of CHF re-hospitalization or death from any cause within six months.

The study found no difference in the primary endpoint between placebo (17.9%) and the pooled adenosine dose groups (16.3%) nor, with the single adenosine groups separately appraised (low and high adenosine dose 16.5% vs. 16.1%, respectively, *p* = 0.43). A prespecified sub-study, including 243 patients in which infarct size was measured by technetium-99m sestamibi tomography, observed a trend toward a smaller median infarct size with adenosine than the placebo group, whereas median infarct size was significantly reduced in the high-dose adenosine group only.

Hence, taken together, adenosine infusion after AMI appeared to reduce infarct size with a higher adenosine dosage in anterior AMI patients, but did not improve clinical outcomes of death for all causes and re-hospitalization for CHF.

Other more recent studies in humans provided contrasting results regarding the impact of adenosine administration on myocardial infarction size [174,175,176,177], whereas its impact on all-cause mortality and myocardial infarction remained neutral in meta-analyses [178].

Adenosine is currently part of the armamentarium of the interventional cardiologist and is commonly used as an intracoronary bolus in case of flow disturbances including slow-flow or no-reflow after stenting but is not recommended for routine use in case of no flow disturbances. Reconcile the current evidence from multiple studies of adenosine in AMI is difficult, yet several explanations could justify, at least in part, the contrast between a lack of clinical benefit of adenosine in trials and the premises of pre-clinical studies (Figure 3). Adenosine has a very short half-life, with rapid metabolization in the bloodstream, hence several variables of adenosine administration are of paramount importance: the route of administration, namely intravenous or intracoronary, has a great potential impact to provide an effect during MI. In theory, intracoronary administration may provide additional benefit being close to the tissue in which the metabolic changes of AMI are occurring.

For the same reason, another key point is the duration of infusion. Given the very short half-life of adenosine, it is unlikely that a short infusion, or a bolus, could give a definite impact on the disease. In fact, animal studies provided evidence of efficacy for a prolonged infusion but not for a single bolus in a swine model of AMI [179]. Moreover, the drug dosage is of critical importance. As seen in the AMISTAD-II trial, a high dose infusion has proven useful in reducing infarct size, whereas a lower dose was not [173]. The importance of adenosine dose has been confirmed also in other studies, highlighting that selecting a too-low dose of adenosine is not useful in the AMI stetting and could even be detrimental: in fact, while high concentrations of adenosine limited neutrophil recruitment (via A_2A_ and A_2B_R subtypes receptors), lower local levels of adenosine have been in turn noted to promote neutrophil recruitment (via A_1_ and A_3_R subtypes) [180].

In conclusion, whether the benefit of routine adenosine use in AMI might transfer in a significant improvement of clinical outcomes probably depends on the intersection between drug dose/duration and the timing of clinical presentation. Future study designs should focus on these pitfalls in the hope of filling the gap between bench and long-term clinical benefit.

Drugs that increase adenosine levels: While the effects of adenosine on AMI are controversial, other drugs can indirectly increase adenosine levels and have been studied in cardiovascular clinical trials. Dipyridamole, an antiplatelet agent, acts as a nucleoside transport inhibitor that blocks the cellular reuptake of adenosine, thus increasing extracellular adenosine levels. This has been tested in the Persantine–Aspirin Reinfarction Study (PARIS) which examined the efficacy of dipyridamole at the dose of 75 mg in combination with aspirin versus aspirin alone or placebo in 2026 patients with a prior myocardial infarction (8 weeks to 5 years before enrolment) for an average follow-up of 41 months. Ultimately, no statistically significant difference was observed between treatment with dipyridamole plus aspirin vs. aspirin alone. It has to be highlighted that PARIS trial was performed in a period in which current revascularization procedures were not available.

Adenosine Antagonists: In contrast with the abundance of pre-clinical and clinical evidence of adenosine adjunctive therapy in AMI, there is a paucity of data regarding the impact of adenosine receptor antagonists in this setting. Abundant indirect evidence in this field could be obtained from dietary elements that are known to inhibit adenosine receptors such as caffeine.

Caffeine, which belongs to the class of methylxanthines, is the most commonly used neuro-stimulant worldwide, typically through coffee or tea consumption. The most prominent mechanism of action of caffeine is its reversible blockage of all adenosine receptors subtypes (A1, A2A, A2B, and A3), although presenting the highest affinity for the A2A receptor.

The concerns regarding the possible detrimental effects of caffeine on cardiovascular health have been extensively studied.

Freedman et al. evaluated the association of coffee drinking with different causes of mortality in a large, prospective cohort of more than 400,000 individuals of 50 to 71 years of age and without prior history of cancer, heart disease, or stroke at the time of enrolment [181]. Enrolled individuals have than been prospectively followed for an average of 14 years. After adjustment for potential confounders as tobacco smoking, higher alcohol and red-meat consumption, that were more commonly associated with coffee use, a significant inverse association between coffee consumption and total mortality was observed. The association between lower mortality and coffee consumption was evident in both men and women, and was dose-dependent, being consistently lower among those consuming one cup and up to 6 cups of coffee per day. Men and women drinking 6 or more cups of coffee per day had a 10% and 15% lower risk of death, respectively. Importantly, coffee drinking was associated to a reduced incidence of mortality due to heart disease.

Similarly, Mukamal et al. evaluated the association of tea consumption before an AMI with survival afterwards [182]. Authors evaluated 1900 patients hospitalized with a confirmed AMI collecting self-reported usual weekly caffeinated tea consumption during the year before AMI using a standardized questionnaire. Compared with non-drinkers, adjusted mortality was lower among moderate tea drinkers (14 cups per week) (HR 0.69; 95% CI, 0.53 to 0.89) and heavy tea drinkers (≥14 cups per week) (HR 0.61; 95% CI, 0.42 to 0.86). The association of tea and mortality was similar for total and cardiovascular mortality. Yet, while evaluating these indirect evidences it has to be highlighted that effects of coffee and tea could or could not be related to caffeine. Coffee and tea contain more than 1000 compounds that might affect health, and other elements including polyphenols and flavonoids have antioxidant effects that can have a potential impact on mortality. In fact, in analyses stratified according to the predominant type of coffee/tea consumed (caffeinated vs. de-caffeinated), the association with total mortality appeared to be similar [181,182].

Alternatively, we could speculate that caffeine could have a direct cardioprotective effect. In fact, while adenosine might be protective by stimulating coronary vasodilatation and increased coronary flow, long-term adenosine exposure may be deleterious and may elicit chronic inflammation and organ damage which may be modulated by adenosine blockage though caffeine [183]. On this matter also the level of adenosine might be relevant, as low and sustained local levels of adenosine have been noted to promote neutrophil recruitment and inflammation [180]. Finally, it is possible that caffeine and adenosine might have a synergistic effect in cardioprotection. Conley et al. found that rats exposed to increasing doses of caffeine or the non-selective adenosine antagonists 8-p-sulphophenyltheophylline showed a significant dose-related, saturable increase of adenosine plasma levels [184]. Varani et al. demonstrated a significant upregulation of adenosine A2A receptors in human platelets after caffeine treatment, with an increase of density and affinity in platelet membranes. This upregulation lead to a 2–3 fold increase of cAMP levels and a significant inhibition of platelet aggregation after subsequent stimulation with A2A receptor agonists [185]. Hence, while speculative, it is conceivable that mild chronic caffeine consumption may modulate adenosine levels, may upregulate adenosine receptors and may reduce platelet aggregability which might explain the epidemiological evidence of better outcomes in moderate coffee/tea drinkers [181,182]. These potential cardioprotective effect could be elicited especially during coffee-free periods of the day (e.g., night-time).

In conclusion, while it is unknown whether caffeine and its adenosine blockage activity or other active compounds present in coffee and tea provide protection, observational studies support the association between coffee/tea drinking and improved survival in patients after AMI.

## 8. Conclusions

Several findings consistently demonstrated that Wnt/β-catenin pathway is involved in the pathogenesis of myocardial damage in post-MI patients, by stimulating scaring and ventricular chamber dilatation. Few studies focused on the interactions between these networks. In this review, we identified a possible interlink between adenosine system, purinergic system and the downstream Wnt/β-catenin pathway, which may promote the progression of myocardial injury up to heart failure when this sophisticated balance is not preserved. Therefore, both Wnt inhibitors and therapeutic approaches acting on adenosine and/or purinergic system pathway inhibition may represent a valid cardioprotective strategy for the treatment of heart injury. Yet, future investigations are required to clarify the precise role of these pathways after myocardial infarction.

## Figures and Tables

**Figure 1 biomedicines-09-00204-f001:**
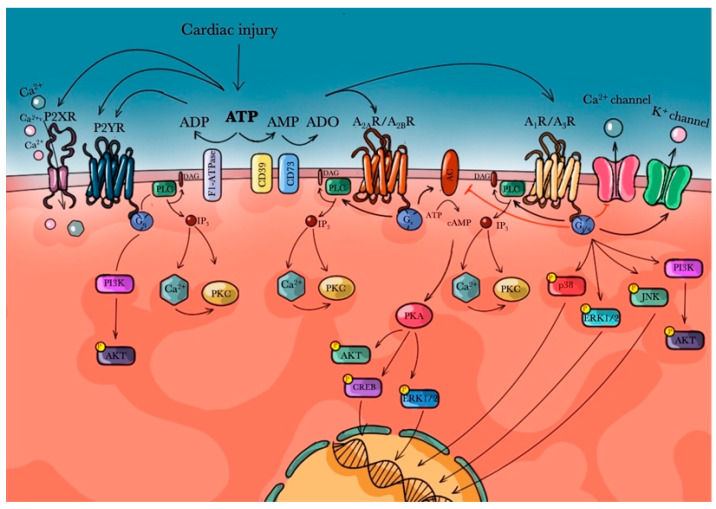
Adenosine receptors role after cardiac injury.

**Figure 2 biomedicines-09-00204-f002:**
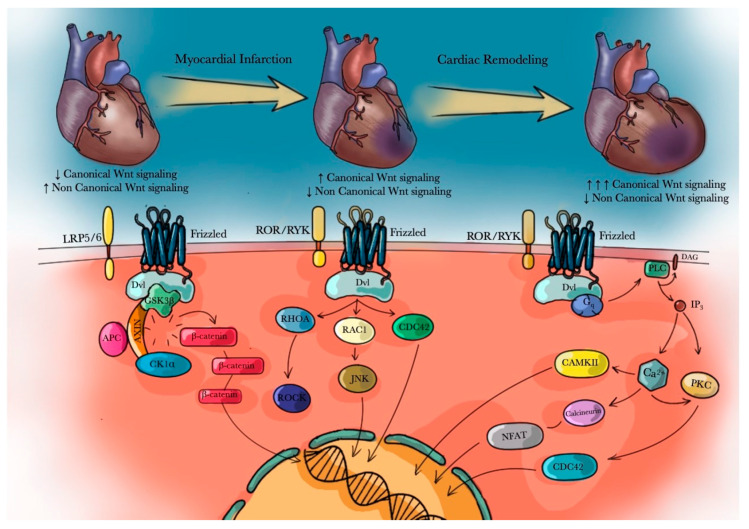
Schematic summary of canonical and non-canonical Wnt/βcatenin signaling pathways activation in both mature healthy heart and under pathological conditions**.**

**Figure 3 biomedicines-09-00204-f003:**
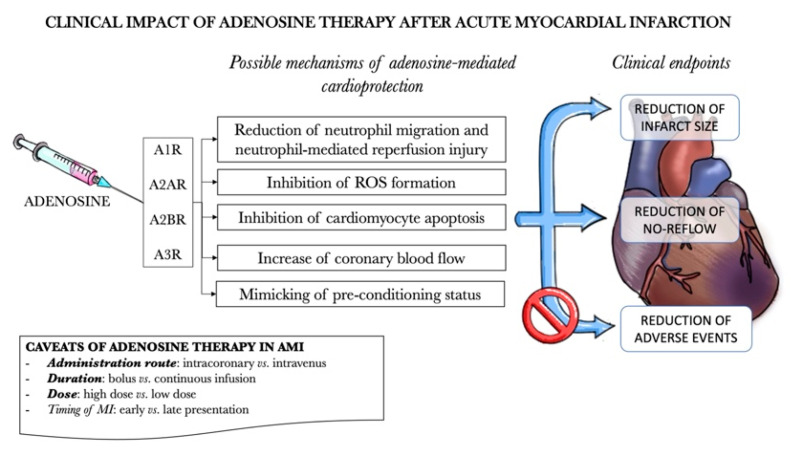
Potential role of adenosine therapy after acute myocardial infarction.

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
