# Peer review of "Role of Adenosine and Purinergic Receptors in Myocardial Infarction: Focus on Different Signal Transduction Pathways"

_biomedicines, 2021, doi:10.3390/biomedicines9020204_

Round 1

Reviewer 1 Report

The authors have improved the manuscript and answered all the questions adequately

I have no more comments

Reviewer 2 Report

The authors have addressed all my concerns effectively. Good work.

This manuscript is a resubmission of an earlier submission. The following is a list of the peer review reports and author responses from that submission.

Round 1

Reviewer 1 Report

The authors aim to review the role of adenosine and its receptors on myocardial ischemia protection, with a special focus on the implication of Wnt. beta catenin pathway.

The study is of interest and well writen.

I have however some major concern

The part 3 should be reversed with part 2 because adenosine receptors are purinergic receptor subtypes. Thus the part 3 is more general than part 2 and these parts should be reversed.

Generally the precise link between adenosine receptor activation  and the Wnt/ beta -catenin signaling pathway is poorly advocated. While the transduction signal pathways that follow adenosine receptors  activation is well done (part 2), Please precise in 4-5 lines why the authors suspect that the acitvation of this precise cascade is associated with the MI protection in comparison with other trasduction signal pathways. It remains confuse as presented yet. Furthermore if the possible therapeutic role of Wnt inhibitor in the prevention of fibrosis is advocated, a possible role of adenosine receptor agonist is not advocated (only infusion of adenosine  and drugs that increase adenosine plasma level were advocated; see ATL 313 a selective A2a adenosine receptor agonist reduces myocardial infarct size ; Wangde Dai et al The open cadiovascular Medicine Journal 2021). Authors should add a short pargarph in this perspective.

Furtheremore the title is not adequat since the focus on Wnt/ beta catenin pathway is poorly advocated. I suggest : Role of adenosine and purinergic receptors in myocardial infarction: Focus on different signal transduction pathways.

Minor

Part 2:

a recent review on coronary artery disease and adenosine should be cited:

Paganelli F et al Cardiovasc Res 2020

.. to the class of nucleotidases: the ectonucleotidases CD 39 and CD 73

Part 3

P2 receptors desentizitazion results in a negative ionotropic effects

What do you mean?

Part 6

AMISTAD II: i think its is 50µG and 70µG/Kg/min and not mG.

Concerning the paradoxal effects of coffee on and mortality; i think it coulb be advocated two explanations.

First if acute high adenosine level may be protective via coronary vasodilation, long term exposure to high adenosine may be deleterious (see Borea P Br J Pharmacol 2017); Second  caffeine induces an up regulation of both A2a R (Varani K Circulation 1999) and adenosine level (LA Conley, Nature 1997). Thus while caffeine has 4-5 hours half life, this up regulation of adenosine system occurs several hours and may protect the myocardium specially during the night when subjects are caffein free.

Author Response

As suggested by the Reviewer part 3 and part 2 were reversed to ameliorate the sequence of the text. Thank you for your suggestion.

As requested by the Reviewer we specified the possible relationship of Wnt pathway with MI. In particular, canonical Wnt pathway activation might be enhanced following myocardial injury due to mechanisms of "counter-regulation" of the infarct process, in order to preserve the heart attacked area. However, the activation of this pathway in a post-infarct environment could contribute to the establishment of LV remodeling. An excessive canonical Wnt activation might activate signaling cascades, promoting rearrangements of the cytoskeletal structure of cardiomyocytes. This final effect could potentially lead to cardiac remodeling and let us to hypothesize that abnormalities of these path-ways might be considered as risk factors for the development of different cardiovascular pathological conditions.

In agreement with the Reviewer suggestions, we provided an additional paragraph on adenosine receptors, thus including the recommended references.

As suggested by the Reviewer we modified the title. Thank you for your suggestion.

As requested by the Reviewer we added the reference.

As requested by the Reviewer, we explained that P2 receptors desensitization, resulting in a negative ionotropic effect, can occur with increased ATP levels, thus weakening the force of muscular contractions of cardiomyocytes and inducing dysfunctions in myocardium.

In agreement with the Reviewer correction, we changed the doses (50μg and 70μg/Kg/min).

We thank the reviewer for these precious insights, and we read with interest the references suggested. We agree that the activation or blockage of the adenosine pathway in patients with prior myocardial infarction is still controversial, and we appreciate the Reviewer suggestion to improve our current version of the manuscript. We have now updated the text including some of the Reviewer suggestions as follows:

“Alternatively, we could speculate that caffeine could have a direct cardioprotective effect. In fact, while adenosine might be protective by stimulating coronary vasodilatation and increased coronary flow, long-term adenosine exposure may be deleterious and may elicit chronic inflammation and organ damage which may be modulated by adenosine blockage though caffeine. On this matter also the level of adenosine might be relevant, as low and sustained local levels of adenosine have been noted to promote neutrophil recruitment and inflammation). Finally, it is possible that caffeine and adenosine might have a synergistic effect in cardioprotection. Conley et al. found that rats exposed to increasing doses of caffeine or the non-selective adenosine antagonists 8-p-sulphophenyltheophylline showed a significant dose-related, saturable increase of adenosine plasma level. Varani et al. demonstrated a significant upregulation of adenosine A2A receptors in human platelets after caffeine treatment, with an increase of density and affinity in platelet membranes. This upregulation lead to a 2-3 fold increase of cAMP levels and a significant inhibition of platelet aggregation after subsequent stimulation with A2A receptor agonists. Hence, while speculative, it is conceavable that mild chronic caffeine consumption may modulate adenosine levels, may upregulate adenosine receptors and may reduce platelet aggregability which might explain the epidemiological evidence of better outcomes in moderate coffee/tea drinkers. These potential cardioprotective effect could be elicited especially during coffee-free periods of the day (e.g. night-time).”

Reviewer 2 Report

The authors present a well-written and comprehensive review on the role of adenosine and purinergic receptors in AMI with special emphasis on Wnt/Beta-Catenin signaling pathway. Authors discuss molecular ways to mitigate cardiac fibrosis and improvement of LV systolic function post-MI by inhibiting Wnt signaling pathway that is implicated in adverse LV remodeling and heart failure following MI.

The review is broken down in sections explaining adenosine receptors, purinergic receptors physiology and Wnt pathways with following section on Wnt pathway inhibitors. There are some minor comments:

  1. Given the neutral results of AMISTAD and AMISTAD II studies, what is the rationale of authors for adenosine potential integration in future clinical practice? What is the future outlook of adenosine agonists and/or adenosine?
  2. Are there any human studies with Wnt pathway inhibition. Clinical outlook for this approach should be discussed more extensively.
  3. Toxicity of potential Wnt pathway inhibiton should be discussed. What are potential perils and pitfalls of this approach?
  4. A graph depicting Wnt/beta-catenin pathway would be welcome (figure) in the context of MI pathophysiology.

Author Response

We thank the reviewer and this point is very well taken. Unfortunately, the results of AMISTAD and AMISTAD II could not definetely confirm an advantage of routine use of adenosine in the setting of acute MI that was hypothesized in many pre-clinical and clinical studies highlighting a reduction of infarct size and microvascular obstruction.

However, as we pointed out in the text and in Figure 3 many factors make adenosin difficult to test in the clinical setting, and only a study design taking into account these variables could hope to reconcile these inconsistencies.

We have now better clarified in the text the current use, the potential pitfalls and the future outlook of adenosine trials in the text as suggested by the Reviewer:

Adenosine is currently part of the armamentarium of the interventional cardiologist and is commonly used as an intracoronary bolus in case of flow disturbances including slow-flow or no-reflow after stenting but is not recommended for routine use in case of no flow disturbances. Reconcile the current evidence from multiple studies of adenosine in AMI is difficult, yet several explanations could justify, at least in part, the contrast between a lack of clinical benefit of adenosine in trials and the premises of pre-clinical studies (Figure 3).” AND “In conclusion, whether the benefit of routine adenosine use in AMI might transfer in a significant improvement of clinical outcomes probably depends on the intersection between drug dose/duration and the timing of clinical presentation. Future study designs should focus on these pitfalls in the hope of filling the gap between bench and long-term clinical benefit.

In agreement with the Reviewer suggestion, we discussed about some Wnt inhibitors used in clinical trials such as WNT974, ETC-159, RXC004, CGX1321, Vantictumab and Ipafricept.

We highlighted that Wnt inhibitors might be considered as a possible therapeutic strategy, in particular in association with bone protective agents to reduce side effects. In fact, Wnt inhibitors use might be related to the appearance of side effects such as bone mass impairment and bone resorption increase. The potential clinical application needs to be deeply evaluated: further studies should investigate the sophisticated interaction of Wnt signaling with other pathways and the side effects/toxicity related to their use.

As suggested by the Reviewer we modified the figure. Wnt signaling plays a pivotal role in several developmental processes and its modulation is also essential for cardiovascular physiology. In the mature heart the canonical Wnt/βcatenin signaling is significantly reduced to inhibit an excessive heart development after cardiomyogenesis, thanks to the antagonizing activity of the non-canonical Wnt/βcatenin pathway. MI may induce the activation of several mediators which promote the canonical pathway stimulation, which is further enhanced during cardiac remodeling, due to the effects of the post-infarct environment. In the figure, the downstream molecules recruited after the activation of the canonical Wnt/β-catenin signaling pathway and the non-canonical Wnt/Planar Cell Polarity (PCP) pathway and WNT/Ca2+ pathway, respectively, were drawn.